# Integrated molecular diode as 10 MHz half-wave rectifier based on an organic nanostructure heterojunction

Tianming Li [1,2,3,6], Vineeth Kumar Bandari [1,2,3,6], Martin Hantusch[4], Jianhui Xin[5], Robert Kuhrt [4], Rachappa Ravishankar[1,2], Longqian Xu[1,2], Jidong Zhang[5], Martin Knupfer[4], Feng Zhu [1,2,3,5✉], Donghang Yan[5] & Oliver G. Schmidt[1,2,3]

Considerable efforts have been made to realize nanoscale diodes based on single molecules or molecular ensembles for implementing the concept of molecular electronics. However, so far, functional molecular diodes have only been demonstrated in the very low alternating current frequency regime, which is partially due to their extremely low conductance and the poor degree of device integration. Here, we report about fully integrated rectifiers with microtubular soft-contacts, which are based on a molecularly thin organic heterojunction and are able to convert alternating current with a frequency of up to 10 MHz. The unidirectional current behavior of our devices originates mainly from the intrinsically different surfaces of the bottom planar and top microtubular Au electrodes while the excellent high frequency response benefits from the charge accumulation in the phthalocyanine molecular heterojunction, which not only improves the charge injection but also increases the carrier density.

[1] Material Systems for Nanoelectronics, Chemnitz University of Technology, 09107 Chemnitz, Germany. [2] Institute for Integrative Nanosciences, Leibniz IFW Dresden, 01069 Dresden, Germany. [3] Center for Materials, Architectures and Integration of Nanomembranes (MAIN), Chemnitz University of Technology, 09126 Chemnitz, Germany. [4] Institute for Solid State Research, Leibniz IFW Dresden, 01069 Dresden, Germany. [5] State Key Laboratory of Polymer Physics and Chemistry, Changchun Institute of Applied Chemistry, Chinese Academy of Sciences, Changchun 130022, China. [6]These authors contributed equally: Tianming Li, Vineeth Kumar Bandari. ✉email: zhufeng@ciac.ac.cn

Over the past decades, molecular-scale electronics have received considerable attention due to the wide tuning range of the electrical properties and the greatly reduced size in comparison with traditional inorganic electronic devices[1]. Molecular rectifying diodes as proposed by Aviram and Ratner[2] are the archetypal molecular devices that are expected to convert high-frequency alternating current (AC) to direct current (DC) and provide low-cost solutions for potential electronic applications, such as the widespread radio-frequency identification tags[3]. Generally, if the leakage current is negligible, the maximum operating frequency ($f_{MAX}$) of a diode-based rectifier is given by $f_{MAX} \propto \mu/L^2$, where μ and $L$ are the mobility and thickness of the semiconductor spacer, respectively[4]. Therefore, assuming nanometer thickness and high charge transport capability, it is predicted that ultrathin organic ensembles and self-assembled monolayers (SAMs) could achieve a rectification frequency in the terahertz range[5]. GHz and THz operation has already been demonstrated in prototypical molecular systems. For example, a scanning microwave microscope tip measured a rectification ratio of 4 dB in the reflection coefficient $|S_{11}|$ of a molecular diode operating at 17.8 GHz[6]. Plasmon quantum resonances controlled by molecular tunnel junctions have even been observed at THz frequencies[7]. However, integrated molecular rectifier devices that convert high-frequency AC input into DC output have not yet been reported.

Because of the fragile nature of ultrathin organic/molecular materials, any attempt to deposit top electrode materials directly by using established physical vapor deposition methods could lead to critical damage and short circuits caused by metal penetration. Therefore, thick organic spacers have been introduced (tens to hundreds of nanometers) to avoid any shortcuts, and these "thick" organic devices have delivered rectification up to the gigahertz frequency range[4,8]. In order to realize nondestructive electric contacts to molecular ensembles and ultrathin organic thin films, various techniques have emerged for creating nanogapped electrodes for molecular devices, including break junctions[9], crossed wire junctions[10], liquid oxide/metals (for example, $Ga_2O_3$/EGaIn[11]), and scanning tunneling microscopy (STM)/atomic force microscopy (AFM) tips[5,6]. However, these techniques are unable to realize functional molecular rectifiers operating at high frequency with an integrated top electrode.

Despite the success in efficiently controlling the rectification ratio ($10^2$–$10^5$) by varying chain length of SAMs[12] and selecting different bottom electrode materials (Au, Ag, or Pt)[13], state-of-the-art frequency performance has remained far from theoretical expectations. Nijhuis et al. reported a molecular half-wave rectifier consisting of $Ag^{TS}$-$SC_{11}Fc_2$//$Ga_2O_3$/EGaIn[14]. A low retention of 18% of the input voltage amplitude was achieved at 50 Hz. This is the reported state-of-the-art molecular-scale rectifier device that possesses evident frequency characteristics and can convert AC into DC. After decades of efforts, molecular junctions with a good rectification ratio under DC have been realized. However, these devices fail to deliver rectification at high-frequency AC. One reason is that even at forward bias the resistances of the molecular diodes are too large (>$10^6$ Ω)[15]. The large resistance not only leads to very small output voltage retention but also produces a lot of heat, which reduces the lifetime of the devices (only several tens of minutes)[14,15]. Therefore, tuning the local electronic structures and increasing the charge transport capability of the ultrathin organic layers are very important.

In this report, we demonstrate fully integrated molecular-scale rectifiers based on ultrathin molecular junctions that are capable of working at more than 10 MHz. The excellent frequency performance benefits from a soft top contact created by a rolled-up metal microtube and charge transfer between Au electrode and the phthalocyanine molecular heterojunction. By employing rolled-up technology, strain layers (Au/Ti/Cr) provide a damage-free and robust contact to the molecular layers, which allows us to decrease the active layer thickness ($L$) to only a few nanometers. The rectifying behavior originates from the difference in the interface properties of the bottom planar and top microtubular Au contacts, which in turn is caused by the different fabrication processes. On the other hand, the high-frequency performance is due to the improvement in charge injection and transport from the bottom Au electrode to the organic layers, which is achieved by charge accumulation in the phthalocyanine molecular heterojunction. This is the first time a nanometer-thick integrated organic rectifier device has been created with operation frequencies beyond 1 MHz.

## Results

**Construction of organic heterojunction.** One of the most important prerequisites for integrated devices is the stability, especially for organic materials that must be maintained throughout the complicated fabrication process and the subsequent long-term operation lifetime. Copper phthalocyanine (CuPc) has stood out for its exceptional thermal and chemical stability since it was firstly synthesized early last century[16]. In our work, two critical obstacles have to be overcome before an ultrathin CuPc molecular layer could be applied as the organic semiconducting spacer in the molecular-scale devices; these are the poor conductivity and inefficient charge injection from the metal electrodes. These two factors lead to a large voltage drop over the diodes resulting in low outputs of the rectifiers[14]. There are at least two strategies for improving the electrical performance of CuPc: first, control the orientation of the planer molecules by modifying the surface of the substrate[8,17], and second, rearrange the carrier distribution by introducing a heterojunction or dopants[18,19]. Previously, heterojunctions between metal phthalocyanines (MPcs) and fluorinated MPcs (F-MPcs) have been employed to enhance the carrier concentration by hybridization at the interface[20,21]. Therefore, in this work, 1 nm fluorinated cobalt phthalocyanine ($F_{16}CoPc$) was introduced between the Au substrate and the nanometer-thick CuPc layer to act both as the buffer layer and n-type semiconductor. Grazing incidence X-ray diffraction (GIXRD) was performed to determine the structure of CuPc thin films grown on the Au substrate with and without $F_{16}CoPc$, as shown in Fig. 1 and Supplementary Fig. 1. Firstly, no obvious diffraction peak was detected with the deposition of 1 nm $F_{16}CoPc$ on the Au substrate (Fig. 1a), indicating that the $F_{16}CoPc$ film is too thin to be detected. The GIXRD patterns for both samples, without (Fig. 1b) and with (Fig. 1c) $F_{16}CoPc$, show the characteristic feature of the polycrystalline CuPc structure, identified by a typical reflection peak at $2\theta \approx 6.9°$ (Supplementary Fig. 1a), corresponding to the (001) lattice plane of α-phase CuPc[22]. This peak arises from the interlayer spacing of tilted molecular stacks. The CuPc molecules grown on the $F_{16}CoPc$ layer exhibit a sharper (001) peak, compared with the growth on bare Au, which implies that the insertion of $F_{16}CoPc$ increases the CuPc molecules' crystallinity, as illustrated in Fig. 1d. Furthermore, in Supplementary Fig. 1a, the additional peaks ($22° < 2\theta < 28°$) of CuPc without $F_{16}CoPc$ were identified as (241), (412), (242), and (250), respectively[16]. This further indicates that the CuPc film grown on the bare Au substrate deviates from a strong preferential orientation in the [001] direction, i.e., there is less crystallinity of the CuPc layer without $F_{16}CoPc$. In addition, the (001) peak of α-CuPc appeared in both out-of-plane and in-plane diffraction patterns (see Supplementary Fig. 1). The (001) peak in the out-of-plane pattern is much stronger than that in the in-plane pattern, especially in the case of CuPc grown on $F_{16}CoPc$, indicating that

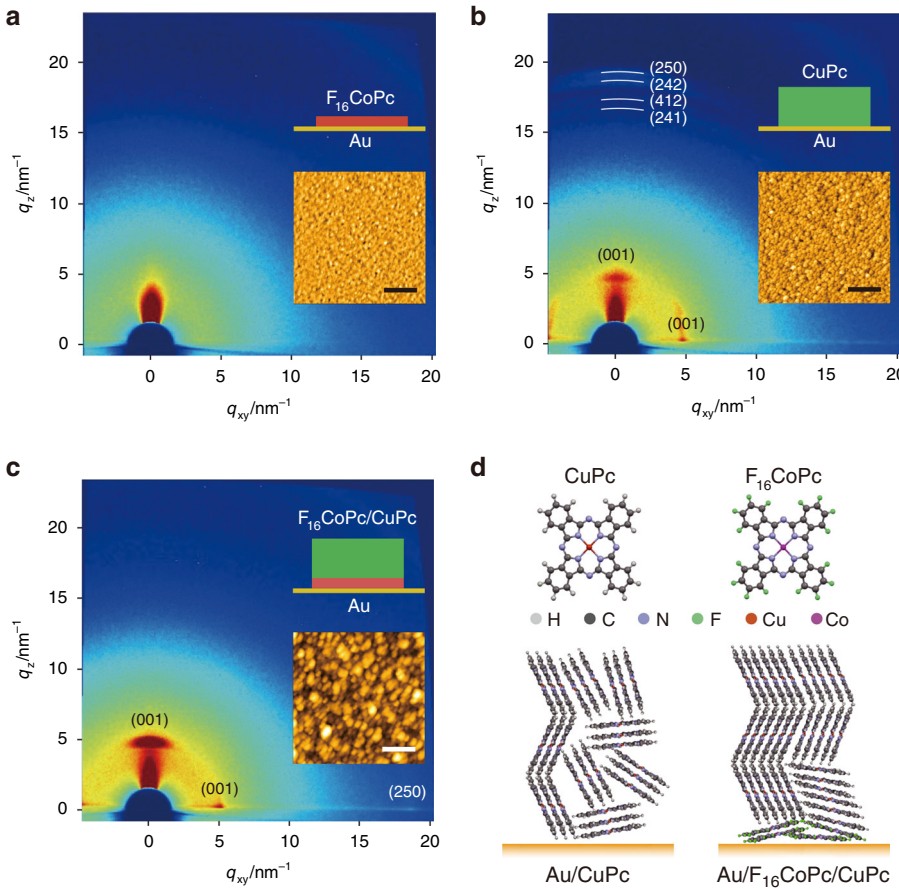

**Fig. 1 Structure and morphology of the phthalocyanine molecule layers. a–c** 2D-GIXRD patterns of $F_{16}CoPc$ (1 nm), CuPc (7 nm), and $F_{16}CoPc$ (1 nm)/ CuPc (7 nm), respectively. The corresponding insets present the layer stacks and AFM images (Scale bars, 250 nm). **d** Schematic molecular packing structures of CuPc grown on bare Au and 1 nm $F_{16}CoPc$ coated Au.

the *ab* planes of most CuPc crystalline domains are parallel to the substrate and a small portion is arranged perpendicular to the substrate. The corresponding AFM images are shown in the insets of Fig. 1a–c, respectively. The organic materials' interfacial and thin film properties are strongly related to the molecule–molecule interactions and the molecule interactions with the underlying substrate. The size distribution of CuPc domains grown on the $F_{16}CoPc$ modified Au substrate is wider (Fig. 1c) in comparison with the CuPc layer grown on the bare Au substrate (Fig. 1a), and larger nanocrystals appear.

Inserting 1 nm $F_{16}CoPc$ not only influences the arrangement of CuPc molecules but also changes the carrier distribution. Ultraviolet photoelectron spectroscopy (UPS) and X-ray photoelectron spectroscopy (XPS) were performed to investigate the electronic structures of the nanoscale heterojunctions. Figure 2a, b shows the evolution of the cut-off energy and the highest occupied molecular orbital (HOMO) during the incremental deposition of the $F_{16}CoPc$/CuPc hybrid layer onto the Cr/Au substrate. The corresponding work function ($\Phi$) and HOMO are summarized in Fig. 2c. It should be pointed out that the Au substrate work function was measured to be 4.18 eV, which is much lower than typical values of clean Au surfaces (~5.0 eV), but corresponds well with values for contaminated Au surfaces[23]. In fact, the Au surface was exposed to air during the substrate transfer to the vacuum chamber, so it was inevitable that contamination by $H_2O$, $NO_2$, $CO_2$, and $O_2$ occurred. This phenomenon was confirmed by XPS of the substrate, as O1s, N1s, and C1s peaks were detected (see Supplementary Fig. 2a).

At the same time, according to the evolution of $\Phi$ shown in Fig. 2c, the energy levels of $F_{16}CoPc$ bend down toward the Au/ $F_{16}CoPc$ interface, while the energy levels of CuPc bend up toward the $F_{16}CoPc$/CuPc interface. In theory, band bending is caused by the charge transfer at the heterojunction[24]. The n-type $F_{16}CoPc$ downward band bending toward the metal–semiconductor interface indicates the electron transfer from the Au to the $F_{16}CoPc$ resulting in an ohmic contact to the molecules because of a charge reservoir residing in the region of the contact[25]. When we attempted to determine the HOMO edges of the ultrathin $F_{16}CoPc$ layers, as shown in Supplementary Fig. 2b, the calculated values for the 0.5 and 1.0 nm $F_{16}CoPc$ layers are 0.21 and 0.16 eV, respectively, which are very close to the work function of the substrate, while the HOMO of bulk $F_{16}CoPc$ is about 1.2 eV[26]. In fact, a typical occupied state arises from the electron transfer from Au to Co3d, which is unique but well known for a $F_{16}CoPc$ monolayer[27]. According to previous reports, the occupied Co state is assigned to the former lowest unoccupied molecular orbital (F-LUMO)[28], which could convert the semiconducting state in ultrathin $F_{16}CoPc$ to metallic states because it is very close to the Fermi level, as shown in Fig. 2c. This charge transfer was further confirmed by the evolution of the Co $2p_{3/2}$ core level at the Au/$F_{16}CoPc$ interface (see Supplementary Fig. 2c). The major peak of the thick $F_{16}CoPc$ film locates at 780.6 eV, which is characteristic for the Co(II) oxidation state. However, in the case of ultrathin $F_{16}CoPc$ films (0.5 and 1.0 nm), another intense peak appears at the binding energy of 778.5 eV, which is attributed to the oxidation state Co(I)[29]. This

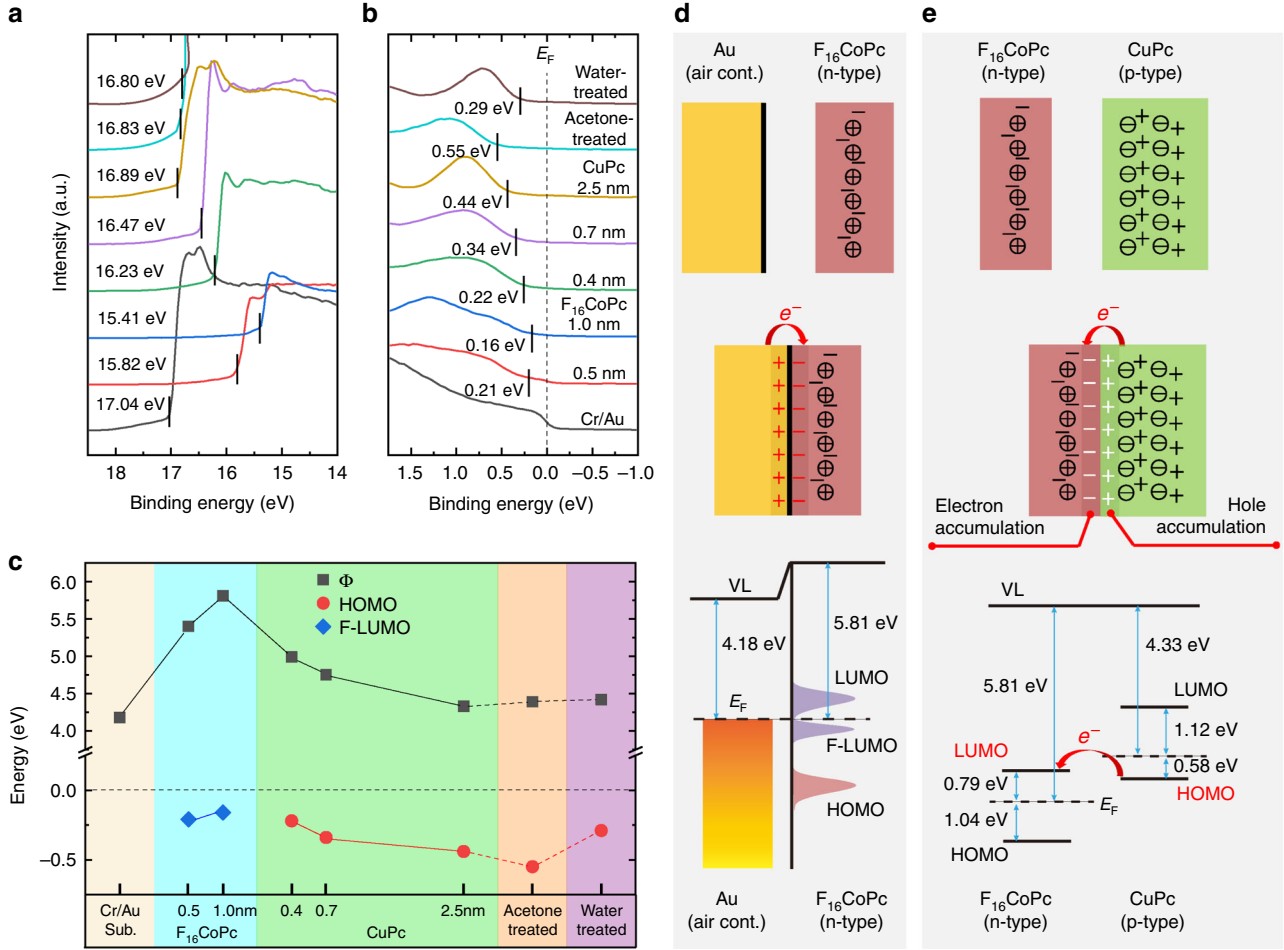

**Fig. 2 Carrier transfer among bottom Au/F$_{16}$CoPc/CuPc. a, b** UPS spectra (He–Iα = 21.22 eV) corresponding to cut-off and HOMO (or VB) regions of the Au/F$_{16}$CoPc/CuPc system. **c** Evolution of work function (Φ) and HOMO peak edges with respect to the Fermi level ($E_F$) of the Cr/Au substrate. **d, e** Schematic diagrams of charge transfer between Au substrate and 1 nm F$_{16}$CoPc, and between n-type F$_{16}$CoPc and p-type CuPc (+/−: free carriers, ⊕/⊖: charge centers).

indicates that some of the cobalt atoms are reduced from Co(II) to Co(I) due to the interfacial charge transfer and consequently a new chemical state, located at lower binding energy, is resolved.

On the other hand, the relative content of Co(I) increased, while the relative content of Co(II) decreased during the deposition of CuPc on 1 nm F$_{16}$CoPc layer, as shown in Supplementary Fig. 2c, d. At the same time, the peak of the Cu 2p$_{3/2}$ core shell from CuPc shifted to higher binding energy, which means that the electron density around Cu decreased (see Supplementary Fig. 2e). Both these two phenomena indicate the electron transfer from CuPc to F$_{16}$CoPc[30]. Therefore, the CuPc energy levels bend up toward the F$_{16}$CoPc/CuPc interface, as mentioned above, in Fig. 2c. In fact, the electron transfer at the F$_{16}$CoPc/CuPc heterojunction can be considered in terms of band structure[20]. As shown in Fig. 2e, the HOMO of CuPc is very close to the LUMO of F$_{16}$CoPc. Once brought into contact, electrons from the p-type CuPc HOMO easily transfer to the n-type F$_{16}$CoPc LUMO leading to hole and electron accumulations in CuPc and F$_{16}$CoPc, respectively. In other words, this is an accumulation heterojunction, very different to the inorganic PN junctions that have a depletion regime. Thus, the inserted, ultrathin F$_{16}$CoPc accepts electrons from both the Au substrate and CuPc resulting in two interfacial regimes where carriers accumulate, as shown in the band alignment diagram (see Supplementary Fig. 3). The heterogeneous distribution of holes on

the CuPc side can be expressed by[31]

$$P(x) = N_v \exp\left[-\frac{E_F - E_v(x)}{k_B T}\right], \qquad (1)$$

where $N_v$ is the effective density of states for holes in the valence band (which is constant for a given material and temperature), $E_F$ is the Fermi level, $E_v(x)$ is the valance band (or HOMO) at position $x$, $k_B$ is the Boltzmann constant, $T$ is the temperature, and $k_B T = 0.02588$ eV when $T = 300$ K. Based on Eq. (1), the hole concentration ratios of $P(x = 0.4\ \text{nm})/P(x = 0.7\ \text{nm})$ and $P(x = 0.4\ \text{nm})/P(x = 2.5\ \text{nm})$ are 103.5 and 4964.2, respectively. This indicates that the hole concentration is significantly increased at the interface regime compared to bulk CuPc, due to the existence of the heterojunction.

**Microfabrication of the molecular diode**. As mentioned above, besides increasing the mobility, another potential approach to enhance the frequency performance of diode-based rectifiers is to shrink the thickness of the organic spacer down to several nanometers, or even to molecular scale. In this work, a soft contact provided by rolled-up nanomembranes is employed to realize rectifier devices based on a molecular-scale organic layer. Briefly, an Au layer is deposited on a finger-shaped mesa structure, which acts as the bottom contact electrode on which the organic layer(s)

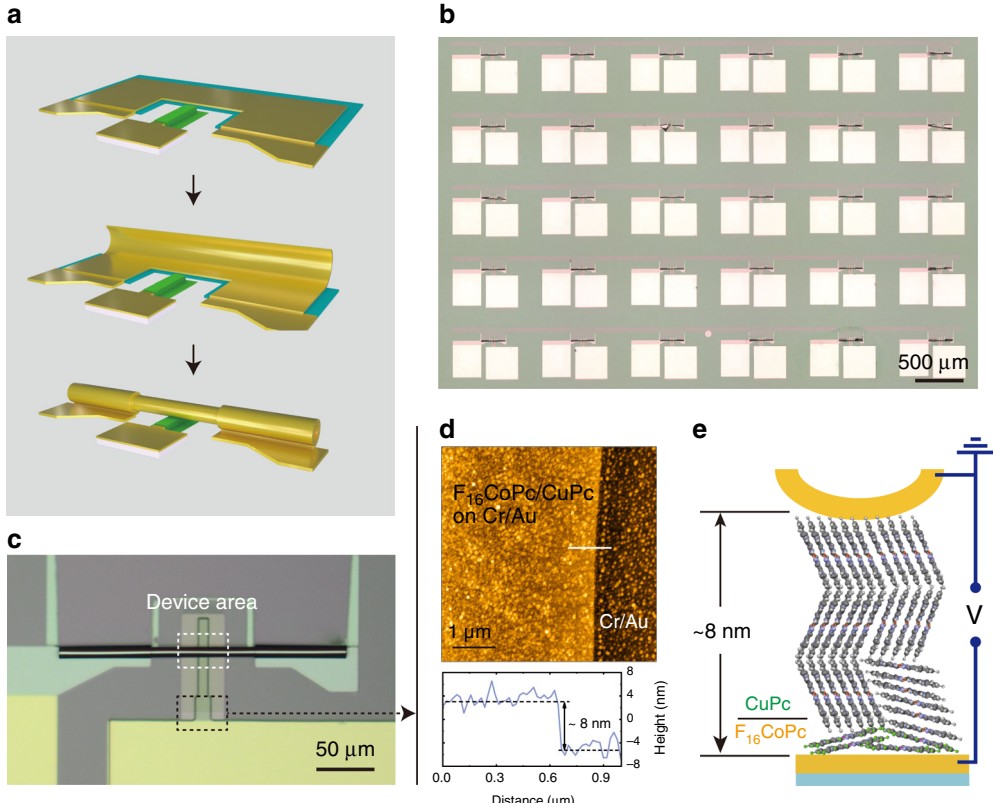

**Fig. 3 Configuration of the molecular-scale rectifiers. a** Formation of rolled-up tube. **b** Micrograph of the diode device array. **c** A typical single device based on rolled-up soft contact. **d** Tapping mode AFM image of $F_{16}$CoPc (1 nm)/CuPc (7 nm) grown on mesa (as marked in **c** by the black dotted box) and corresponding height profile of the AFM image. **e** Conceptual picture of the Au (finger)/$F_{16}$CoPc (1 nm)/CuPc (7 nm)/Au (tube).

are grown. Ge and Au/Ti/Cr nanomembranes are consecutively deposited and patterned as sacrificial layer and strained metallic layers, respectively, whereby the strain layers roll up when the sacrificial layer is selectively etched away by deionized (DI) water. After rolling, the rolled-up metallic nanomembranes provide a damage-free and self-adjusted top electrode for the fragile, ultrathin, organic materials, thus forming a metal/organic/metal sandwich structure (Fig. 3a). A more detailed description of the fabrication process is provided in the "Methods" section (also see Supplementary Fig. 4) and previous reports[32–34]. Figure 3b shows the microscope image of the device array, indicating the feasibility of integration as well as reproducibility, which are key ingredients for practical applications. The rolled-up tubes are formed homogenously and the average diameter is about 10 μm, as shown in Fig. 3c. By combining the soft contact and heterojunction, an excellent type of organic diodes, based on $F_{16}$CoPc (1 nm)/CuPc (7 nm), was successfully integrated onto the silicon wafer, designated as Au (finger)/$F_{16}$CoPc (1 nm)/CuPc (7 nm)/Au (tube). A tapping mode AFM image of $F_{16}$CoPc/CuPc grown on the mesa part (as marked in Fig. 3c by the black dotted box) is shown in Fig. 3d. The corresponding height of the $F_{16}$CoPc/CuPc hybrid layer is 8 ± 2 nm. Compared with the size of the single planer phthalocyanine molecule (~1.5 nm), it is reasonable to claim that the organic spacer, $F_{16}$CoPc/CuPc, falls into molecular scale. Figure 3e shows the concept image of the molecular-scale organic diode in which $F_{16}$CoPc/CuPc heterojunction layer is sandwiched between the Au finger electrode and Au tube electrode.

**Characteristics of the molecular diode**. For the devices based on Au (finger)/$F_{16}$CoPc/CuPc/Au (tube), there is a thickness-related

tradeoff between current density and rectification ratio, as demonstrated in Supplementary Figs. 5 and 6, and Supplementary Note 1. Devices based on an appropriate thickness (8 nm) of the organic hybrid spacer not only have a high rectification ratio but also high forward current density. Moreover, two kinds of mesas with different nominal widths ($W_{design}$), i.e., 5 and 10 μm, are designed. However, the real widths ($W_{real}$) contacting the tube electrodes are smaller, about 1.3 and 7.4 μm, respectively. The shrinking of the designed width is caused by isotropic underetching of the mesas in HF solution (described in Supplementary Fig. 7 and Supplementary Note 2). Both devices with different $W_{design}$ exhibit good rectification ratios and high current densities proportional to their $W_{real}$. Finally, the $F_{16}$CoPc (1 nm)/CuPc (7 nm) hybrid layer with $W_{design} = 10$ μm was chosen to investigate the electrical characteristics. Figure 4a presents the typical DC current-voltage (I–V) characteristics of the molecular diode (denoted with dark cyan squares). The Au tube electrode is held at ground during the measurements while the voltage applied to the Au finger electrode scans from negative to positive. The device shows a good rectification ratio up to 300 at ±2 V. Considering the maximum contact area (7.4 × 10 μm$^2$) a high forward current density of 315 A cm$^{-2}$ at 2 V is achieved (see Supplementary Fig. 8). Furthermore, the devices based on Au (finger)/ $F_{16}$CoPc (1 nm)/CuPc (7 nm)/Au (tube) exhibit excellent stability (see Supplementary Fig. 9 and Supplementary Note 3).

In comparison to Au (finger)/CuPc (7 nm)/Au (tube), devices based on Au (finger)/$F_{16}$CoPc (1 nm)/CuPc (7 nm)/Au (tube) show higher current by more than one order under both forward and reverse bias, while the rectification ratio decreases slightly (see Fig. 4a). This phenomenon is ascribed to the inserting of the ultrathin $F_{16}$CoPc layer that not only improves the CuPc layer's crystallinity but also increases the carrier density. In fact, only a

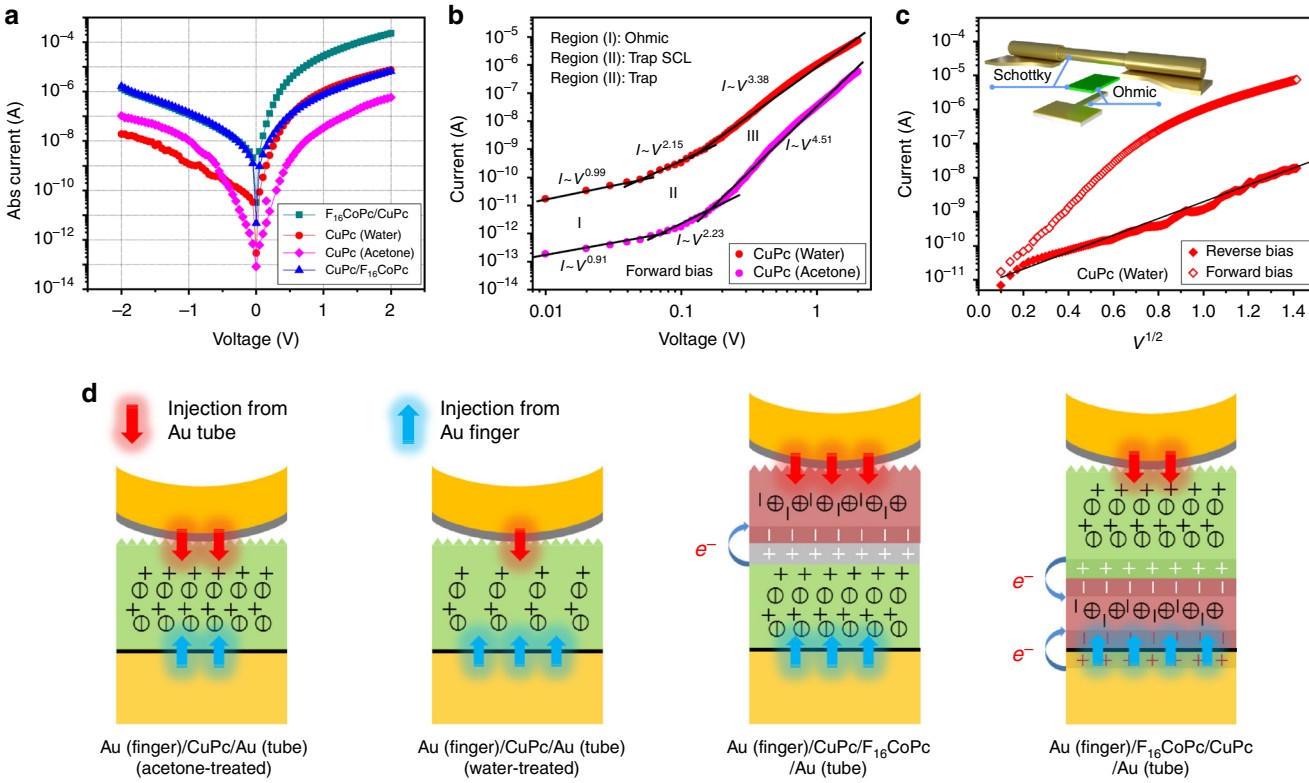

**Fig. 4 Origination of the rectifying behavior. a** $I$–$V$ characteristics of the diodes based on acetone- and water-treated CuPc (7 nm), $F_{16}CoPc$ (1 nm)/CuPc (7 nm), and CuPc (7 nm)/$F_{16}CoPc$ (1 nm), respectively. The mesa widths of all the four cases are 10 μm. **b** Log–log plot of the $I$–$V$ curves of diodes based on acetone- and water-treated Au (finger)/CuPc (7 nm)/Au (tube), showing three regimes distinguished by different $m$ in $I$–$V^m$. **c** log($I$)–$V^{1/2}$ plot of the $I$–$V$ curves of diode based on water-treated Au (finger)/CuPc (7 nm)/Au (tube). The inset illustrates the contacts among bottom finger electrode, CuPc spacer, and top tube electrode. **d** Carrier injection from Au tube and finger influenced by acetone/water treatment, organic heterojunction structure.

very slight rectification ratio of 5 was obtained in a previous report with a similar Au (finger)/CuPc (6.5 nm)/Au (tube) structure[34]. In the fabrication process in the previous work, the CuPc liftoff was performed in acetone. We obtained a similar result by following this process, as shown in Fig. 4a. However, when CuPc was lifted off in water during the process applied in the present work (see Supplementary Fig. 4f–h), the rectification ratio reaches more than 400. In addition, the devices based on water-treated CuPc show a much higher forward current and a lower reverse current compared to the acetone-treated CuPc. Therefore, further questions arise: first, what is the effect of the water and acetone treatments to the molecules? and second, how is the rectifying behavior generated when both bottom and top electrodes are made out of Au?

It can be seen from the double-logarithmic plot shown in Fig. 4b that under forward bias the log($I$)–log($V$) curves of both acetone- and water-treated CuPc show three distinct transport regimes based on the exponent $m$ of $V$, i.e., $I \propto V^m$, the characteristic of which is also known as the space-charge-limited current (SCLC) transport mechanism[35]. One assumption of SCLC is that there is only one type of charge carrier present. For both water- and acetone-treated Au (finger)/CuPc/Au (tube) devices, holes are the carriers. The three regimes are identified as ohmic transport (Regime I), shallow trap-limited SCLC (Regime II), and deep trap-filled limit conduction (Regime III), respectively (see Supplementary Note 4). In other words, transport under forward bias is trap-controlled[36]. It is apparent that there are two transition voltages among the three different conduction regimes described by Supplementary Eqs. (1)–(3). At $V_1$, the injected carrier concentration first exceeds the thermally generated carrier concentration, and the shallow traps are fully

filled at $V_2$[37]. Based on these two critical points, the trap density of acetone- and water-treated CuPc are estimated to be $2.70 \times 10^{18}$ and $1.58 \times 10^{18}$ cm$^{-3}$, respectively. Therefore, the acetone treatment creates more traps compared to the water-treated CuPc leading to lower current. On the other hand, based on the UPS data (see Supplementary Fig. 3), the hole injection barriers of pristine-, acetone-, and water-treated 2.5 nm CuPc with respect to the Au substrate are estimated to be 0.44, 0.55, and 0.29 eV, respectively. Hence, acetone treatment increases the gap between the HOMO of CuPc and the Fermi level ($E_F$) of Au electrodes, but water treatment narrows the gap. In terms of hole concentration in CuPc, the ratios of $P(water)/P(x = 2.5 \text{ nm})$ and $P(acetone)/P(x = 2.5 \text{ nm})$ are 193.09 and 0.02, respectively. In other words, the acetone treatment decreases the hole concentration on the CuPc side, while the water treatment increases it. Apart from the SCLC model (bulk-limited), two possible contact-limited conduction theories (i.e., Fowler–Nordheim tunneling and Schottky emission[38]) are also fitted to clarify the charge carrier injection process from the Au finger to the Au tube electrode (forward bias), as demonstrated in Supplementary Fig. 10 and Supplementary Note 5. The result implies that the transport under forward bias is most likely bulk-limited instead of contact-limited.

Theoretically, SCLC only occurs when the injection electrode forms an ohmic contact, implying that the contact between the Au finger electrode and CuPc are ohmic in both cases[38]. As the devices based on water-treated CuPc show more pronounced rectifying behavior, it is well possible that the contact between the Au tube electrode and the CuPc is of Schottky-type. Because of this, we investigated the $I$–$V$ data of water-treated CuPc with a Schottky effect model (field lowering of the interfacial barrier at the injecting-electrode interface). For standard Schottky emission, the plot of log

($I$) versus $V^{1/2}$ should be linear[39]. As shown in Fig. 4c, the data under reverse bias fit the linear relationship well, while the forward bias data does not. This indicates that the hole injection from the Au tube into the ~7 nm water-treated CuPc ultrathin film complies to the Schottky model[40]. In order to investigate the electronic structure of the top tubular Au electrode, Cr (20 nm)/Ti (15 nm)/Au (5 nm)/Ge (10 nm) multiple layers were deposited onto a silicon wafer to mimic the situation of the strain layers but with a reverse deposition sequence. Before transferring the sample to the XPS and UPS analysis chamber, the Ge layer was removed by using water to expose the Au surface. As a result, the work function of the "Au tube" was measured to be about 4.25 eV (see Supplementary Fig. 11), which is very close to that of the Au finger (4.18 eV). It is very interesting to discover that the symmetric Au electrodes (Au finger and Au tube) result in asymmetric conduction. In fact, the Au (finger)/CuPc and CuPc/Au (tube) interfaces are intrinsically different. CuPc was freshly deposited in ultrahigh vacuum ($10^{-7}$ mbar) onto the air-contaminated Au finger, forming the first interface, while the second interface was formed by the mechanical contact between CuPc and the strained nanomembranes (Au/Ti/Cr), which took place in water. Therefore, the CuPc/Au (tube) interface is more complicated. For example, the element Ge can be observed on the surface of Au (see Supplementary Fig. 11f). In view of the above investigation, the mechanical contact in the integrated device is of Schottky type, which is the intrinsic advantage of the soft contact employed in this work.

Devices based on Au (finger)/CuPc (7 nm)/F$_{16}$CoPc (1 nm)/Au (tube) have also been fabricated. In this case, the CuPc layer is underneath the F$_{16}$CoPc and the device exhibits nearly symmetric $I$–$V$ performance (see Fig. 4a). This phenomenon is interesting and makes sense, as shown in Fig. 4d. In comparison to water-treated Au (finger)/CuPc (7 nm)/Au (tube), the hole injection barrier at the Au finger/CuPc interface is increased for the acetone-treated Au (finger)/CuPc (7 nm)/Au (tube), hence the hole transport across the Au finger/CuPc interface is suppressed, resulting in lower current under forward bias. The slight improvement in carrier injection from the metallic tube may come from modifying the CuPc/Au (tube) interface by acetone. For Au (finger)/CuPc (7 nm)/F$_{16}$CoPc (1 nm)/Au (tube), the introduction of the ultrathin electron-rich F$_{16}$CoPc layer between the CuPc and the Au tube improves the charge transfer across the Au tube resulting in higher current under reverse bias compared to water-treated Au (finger)/CuPc (7 nm)/Au (tube). Therefore, the charge transport in the devices based on the rolled-up soft contact can be effectively controlled by water or acetone treatment and interface modification.

The above discussion implies that the rectifying behavior of Au (finger)/F$_{16}$CoPc/CuPc/Au (tube) originates from the intrinsic difference at the interfaces between Au (finger)/F$_{16}$CoPc and CuPc/Au (tube) rather than the F$_{16}$CoPc/CuPc heterojunction. In theory, there is a built-in potential from CuPc to F$_{16}$CoPc across the F$_{16}$CoPc/CuPc accumulation junction because of the charge transfer effect. For the Au (finger)/F$_{16}$CoPc/CuPc/Au (tube), the built-in potential is in the opposite direction and acts against the forward voltage, while the built-in potential direction is the same and supports the forward voltage for the Au (finger)/CuPc/F$_{16}$CoPc/Au (tube). However, the built-in potential caused by the F$_{16}$CoPc/CuPc junction seems to contribute little to the device rectification, which is consistent with a previous report[21]. This phenomenon is ascribed to the characteristics of the ultrathin MPcs. The 1 nm F$_{16}$CoPc layer changed from semiconducting to metallic due to the charge transfer from the Au substrate. In addition, the thicknesses of the MPc and F-MPc are smaller than the accumulation width (tens of nm)[41], so that the thin hybrid layer tends to be fully accumulated and the junction barrier is negligible. Rectification effects due to the heterojunction interface

would probably be more evident for heterojunctions with much thicker MPc and F-MPc films, which exceed the accumulation width. For example, a diode based on ITO/CuPc (180 nm)/F$_{16}$CuPc (160 nm)/Au exhibited a reverse rectifying characteristic with a ratio of ~20 at ±2 V[41].

**Frequency performance of the molecular rectifier**. The frequency performance of a diode-based organic rectifier is highly dependent on the charge transport capability and thickness of the organic spacer as well as the contact interfaces with the two electrodes[42]. In this work, diodes consisting of several nanometer thin organic films were successfully realized by applying the rolled-up nanotechnology together with the abovementioned organic heterojunction. Both the water-treated Au (finger)/CuPc (7 nm)/Au (tube) and Au (finger)/F$_{16}$CoPc (1 nm)/CuPc (7 nm)/Au (tube) show pronounced unidirectional current behavior. However, the frequency response of the Au (finger)/CuPc (7 nm)/Au (tube) cannot be detected, which may be ascribed to the low conductivity. The frequency characteristics of the diode rectifier based on Au (finger)/F$_{16}$CoPc (1 nm)/CuPc (7 nm)/Au (tube) is promising, as shown in Fig. 5. Figure 5a shows the test setup that was used to characterize the frequency response. With a sinusoidal peak-to-zero input voltage $V_A = 2.5$ V (the root-mean-square (RMS) of the input voltage is 1.77 V), the RMS output voltage $V_{out}$ is about 1.4 V in the low-frequency range. As demonstrated in Fig. 5b, slight leakage occurred in the negative half cycles at 10 kHz, which is attributed to the intrinsic feature of the ultrathin organic layers. At high frequency (100 MHz), the RMS output voltage decreases to 0.4 V, as shown in Fig. 5c. The −3 dB frequency of our diode-based rectifier reaches more than 10 MHz. Compared to previous works, as shown in Fig. 5d, we conclude that it is the first time that fully integrated rectifiers based on nanometer thin organic layers capable of working at high frequency have been achieved[4,8,14,43–50]. The statistical analysis of the as-fabricated device array on chip is provided in Supplementary Fig. 12 and Supplementary Note 6.

## Discussion

Although we can operate our integrated molecular diodes at more than 10 MHz, this frequency is far below the THz range predicted for ideal junctions. The intrinsic cutoff frequency of a diode is given by $f_c = 1/2\pi RC$, where $R = V/I$ is the resistance ($V$ and $I$ are the applied voltage and current, respectively), and $C = \varepsilon_r\varepsilon_0 S/L$ is the planar capacitance ($\varepsilon_r$, $\varepsilon_0$, $S$, and $L$ are the dielectric constant of the material, the dielectric constant of air, the overlapping surface area of the electrodes, and the distance between the electrodes, respectively)[51]. In our Au (finger)/F$_{16}$CoPc (1 nm)/CuPc (7 nm)/Au (tube) configuration, we assume that $S$ of the Au finger and the tube is 74 µm$^2$ and that $\varepsilon_r$ of the F$_{16}$CoPc/CuPc hybrid layer[52,53] is about 4, which yields a capacitance of $3.27 \times 10^{-13}$ F and a theoretical intrinsic cutoff frequency at 2 V of about 0.5 GHz. The discrepancy between experiment and theory is ascribed to the imperfect tube/organic contact at the intricate interface. It is therefore likely that, besides modifying the molecular layers in terms of mobility and thickness, optimizing the contact quality is a feasible strategy to decrease both the contact resistance and the interface capacitance, thus improving the allover performance of the molecular rectifier.

In conclusion, we have realized fully integrated rectifiers based on a molecularly thin organic hybrid layer (F$_{16}$CoPc/CuPc) capable of working at more than 10 MHz. Diode-based rectifiers consisting of an organic spacer with ultrathin thickness have been achieved by applying rolled-up nanotechnology and organic nanostructure heterojunctions. In the present Au (finger)/F$_{16}$CoPc (1 nm)/CuPc (7 nm)/Au (tube) configuration, the

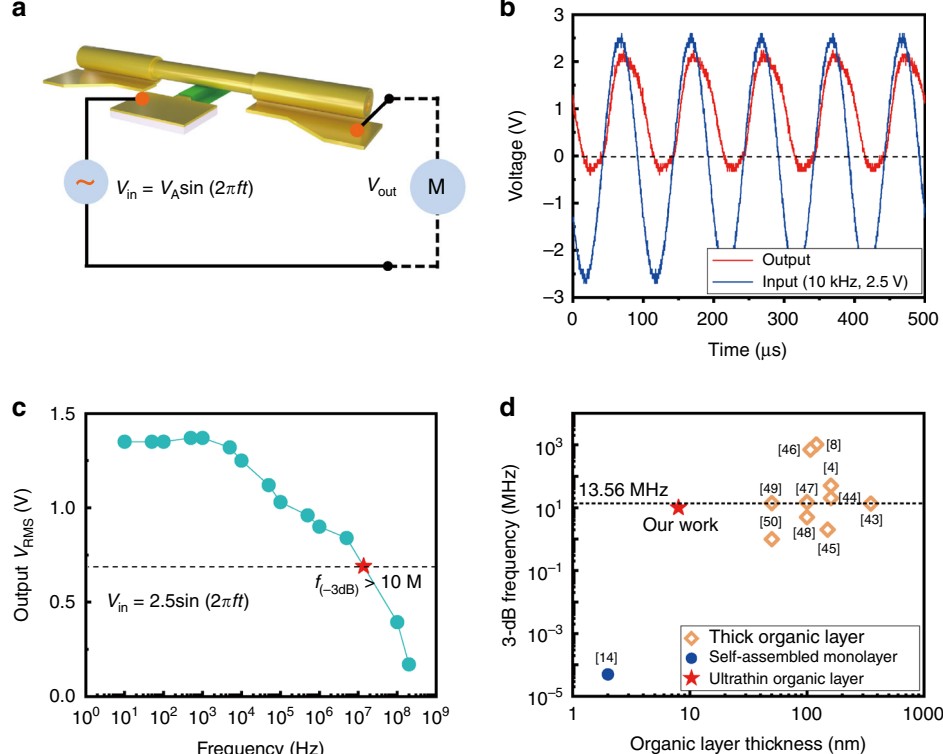

**Fig. 5 Frequency performance of rectifier based on Au (finger)/F$_{16}$CoPc (1 nm)/CuPc (7 nm)/Au (tube). a** Measurement setup for frequency performance. The circuit is open, implying that the resistance of the load is infinite. **b** Rectification behavior of Au (finger)/F$_{16}$CoPc (1 nm)/CuPc (7 nm)/ Au (tube) with $W_{design} = 10\,\mu m$ at 10 kHz. **c** Output DC voltage as a function of the input signal frequency. **d** Comparison of frequency performance between our organic rectifier with previously reported results.

current is increased compared to the single CuPc (7 nm) layer, because the 1 nm F$_{16}$CoPc insert gains electrons from both the bottom Au finger electrode and the CuPc resulting in significant carrier accumulations. The rectifying behavior of the Au (finger)/ F$_{16}$CoPc/CuPc/Au (tube) originates from the fact that the contacts between the Au (finger)/organic and Au (tube)/organic interfaces are intrinsically different (one being ohmic and the other being Schottky type). The integrated molecular diodes proposed here offer the possibility to overcome the tradeoff between good rectification ratio and high operational frequency, and therefore provide an interesting platform for future studies in molecular and organic electronics.

## Methods
**Device fabrication.** The organic rectifiers were fabricated on Si (100) wafer covered by 1 μm SiO$_2$. The configuration of rolled-up contact was realized by relaxing the strained Au/Ti/Cr nanomembrane (see Supplementary Fig. 4). First, the finger-like mesa that would support the bottom electrode was formed by etching ~400 nm of the SiO$_2$ layer in buffered HF solution (Supplementary Fig. 4a). After that, a sacrificial layer of GeO$_x$, oxidized from 20 nm Ge was patterned (Supplementary Fig. 4b), followed by the tri-metallic Au/Ti/Cr nanomembrane (5/15/20 nm), as shown in Supplementary Fig. 4c. The Ti/Cr interface creates the strain gradient and Au layer acts as the top electrode to the organic layer after rolling. Then, an AlO$_x$ layer (10 nm) was formed by atomic layer deposition and patterned for pads deposition and trench opening (Supplementary Fig. 4d). Next in the sequence, the bottom electrode, made of Cr/Au (5/10 nm), was deposited onto the mesa. To enable electrical measurement, a pair of contact pads consisting of Cr/Au (10/50 nm) was patterned, overlapping the area shown in Supplementary Fig. 4e. An ultrathin organic single or hybrid layer was deposited through a MoO$_3$ mask layer prior to the rolling process (Supplementary Fig. 4f). The organic deposition was carried out using low temperature evaporation in vacuum with a base pressure of 10$^{-7}$ mbar. During the deposition, the rate was kept around 0.001 nm/s, and the substrate was maintained at room temperature (Supplementary Fig. 4g). Next, the liftoff was performed in flowing water, instead of acetone. And the wafer was immediately transferred into pure, DI H$_2$O. In this case, the strained metallic nanomembranes curled up because of selective removal of the GeO$_x$ sacrificial layer along the trench. The rolling process stopped at the end of the GeO$_x$ layer (Supplementary Fig. 4h, i). As a result, the rolled-up nanomembranes

contacted the organic ultrathin film from the top, acting as a top electrode without damaging the organic film.

**Device characterization.** The topography images were obtained by a Bruker Atomic Force Microscope with tapping mode. The GIXRD diffraction pattern was taken from Shanghai Synchrotron Radiation Facility on beamline BL14B1 with $k = 1.24$ Å. The $I$–$V$ characteristics were measured at room temperature, using a Keithley 2636A connected to a probe station, with the tube electrode grounded. The frequency characteristics were measured by applying an AC signal to the circuit, as shown in Fig. 5a, using a Tektronix AFG 3252 arbitrary function generator. The output and input signals were monitored using a Tektronix TDS 1002B oscilloscope. The in situ XPS and UPS experiments were carried out using an ultrahigh vacuum system consisting of an analysis chamber interconnected to a preparation chamber. A monochromatic Al Kα source and a He-discharge lamp provided photons with energy of 1486.6 eV for XPS and 21.22 eV for UPS, respectively. The total energy resolution of the spectrometer was about 0.35 eV (XPS) and 0.15 eV (UPS). During UPS measurements, samples were under a bias of 5 V to obtain the correct secondary electron cutoff. F$_{16}$CoPc and CuPc films were grown in steps by thermal evaporation on the Cr/ Au substrate in the preparation chamber and transported to the analysis chamber without breaking vacuum. The film thickness and evaporation rate were monitored and controlled via a quartz crystal microbalance. The actual thickness was further calibrated by monitoring the attenuation in intensity of the Au 4f$_{7/2}$ peak after each deposition.

## Data availability
The authors declare that all data supporting the findings of this study are available from the corresponding author on request.

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

## Acknowledgements

The authors acknowledge Paul Plocica, Martin Bauer, Sandra Nestler, Schröder Liesa, and Ronny Engelhard for their technical support. T.L. acknowledges the support and funding from the China Scholarship Council. V.K.B. acknowledges the support and funding from the European Social Fund. O.G.S. acknowledges financial support by the Leibniz Program of the German Research Foundation (SCHM 1298/26-1). M.K., M.H., and R.K. gratefully acknowledge the financial support by the DFG (KN393/25; KN393/26).

## Author contributions

T.L., V.K.B., F.Z., and O.G.S. conceived the idea. T.L., V.K.B., and F.Z. designed the experiment, and analyzed the data. T.L. and V.K.B. performed devices fabrication and measurements. M.H., R.K., and M.K. carried out UPS and XPS measurements. J.X., J.Z., and D.Y. contributed to the heterojunction design and performed the 2D-GIXRD characterization. R.R. assisted AFM characterization. L.X. contributed to the fabrication process. T.L. and F.Z. wrote the manuscript with input from all authors. F.Z. and O.G.S. supervised the work.

## Competing interests

The authors declare no competing interests.
