## [Peer Review File · Nature Communications]

Reviewers' comments:

Reviewer #1 (Remarks to the Author):

Molecular electronics-based devices have the potential to operate avec very high-frequency, in particular through the THz gap, which is one of the major challenges of electronics today. One advantage of short molecules compared to inorganic counterpart is that the dielectric constant is low (small capacitance), and the conductance can be large due to the nanometric length of the junctions. Although an high-frequency molecular diode (operating at 17 GHz) has already been demonstrated using a scanning microwave microscope tip as a top electrode, it becomes very important to develop such fully-integrated molecular devices operating at high-frequency.

The paper "Fully Integrated High-Frequency Molecular Scale Rectifier Based on Organic Nanostructure Heterojunction" demonstrate such integration for a device operating up to 10 MHz. The rolled-up metal microtube approach is an original way to get a nice top electrical contact in that perspective. The DC electrical properties with rectifications in the range of 300 and current close to mA are nice (Fig.S5). It is a promising demonstration of a fully integrated device towards high-frequency ability (Fig.5b is nice).

I have however couple of issues with the paper in its present form:

- * The Title is misleading as 10 MHz is usually considered as relatively low-frequency by the high-frequency electronics community. Therefore, I suggest that "10 MHz" should replace "High-frequency" as for example: "All printed diode operating at 1.6 GHz", PNAS 111, 11943 (2014)

- * There are many ambiguous references to the literature all along the paper.

To give few examples:

- Abstract: "so far, functional molecular diodes have only been demonstrated in the very low frequency regime". Taking into account Ref.11 ("A 17 GHz molecular rectifier"), "fully integrated molecular diodes" would have been less ambiguous.

=> Similarly for the sentence "This is the first time a nanometer-thick organic rectifier has been created with operation frequency beyond 1 MHz", and the fact that Ref 11 is omitted from the Fig.5d.

- The discussion could be improved in many aspects:

- * It is usually believed by the molecular electronics community that there is a tradeoff in junction area to optimize the density of current (in nanometer-scale junctions, most of the molecule conduct whereas in tens of micrometer junction, only a fraction of the molecules contribute to the current (W. Du et al Nat. Photonics 10, 274 (2016), and that <3 nm junctions should be the best for high-frequency operation as conductance scales typically exponentially with thickness, but the rectification ratio can be degraded. In that context, the present design choices (10 um-wide junction and 8 nm thick film) could have been much better discussed, as well as the capacitive network.

To conclude, the paper clearly demonstrates an integrated molecular device operating at 10 MHz with the response of the device to a sine signal. The rolled-up metal microtube approach was surely efficient to reach this goal and DC properties of the molecular diode are nice. However, given the current litterature results and the proposed title, a 1 GHz demonstration would have been far more impressive, considering refs 6 and 11.

Reviewer #3 (Remarks to the Author):

In this work, Li et al. reported high-frequency (up to 10 MHz) molecular rectifiers based on a

molecularly thin organic heterojunction of F16CoPc/CuPC. They claim that the excellent frequency performance arises from (1) a soft top contact created by a rolled-up Au microtube treated by water and (2) charge transfer between Au electrode and the F16CoPc/CuPC heterojunction.

As this paper reports an interesting and important advance in developing molecular scale rectifiers, it is recommended to be published after addressing the following issues.

(1) In Figure S5a, I-V characteristics of devices with mesa width $W_{\text{mesa}} = 10 \mu\text{m}$ are compared with those with mesa width $W_{\text{mesa}} = 5 \mu\text{m}$. Since mesa width difference is 2, the current difference should be around 2 times if the top contact created by a rolled-up Au microtube is similar. However, it appears that currents are at least 4 or 5 times higher for devices with mesa width $W_{\text{mesa}} = 10 \mu\text{m}$ compared to $W_{\text{mesa}} = 5 \mu\text{m}$. This may suggest that there exists a lateral leakage current. Then it seems that better molecular scale rectifier can be achieved for devices with smaller mesa width. Analyze the effect of lateral leakage current on the rectifier performance.

(2) As the authors mentioned and depicted in Fig. S5b, the real electrode contact area for the device with mesa width $W_{\text{mesa}} = 10 \mu\text{m}$ should be much less than $100 \mu\text{m}^2$. Then a forward current density can be much higher than $\sim 200 \text{ A/cm}^2$ at +2 V. Under such high current across very thin ($\sim 8\text{nm}$) organic layer, one can expect quite high Joule heating, resulting in increased junction temperature. And the effect of Joule heating and junction-temperature rise should depend on the width of mesa. Since the charge carrier injection and conduction depend on the temperature, it is necessary to address the effect of Joule heating on the device I-V characteristics and the rectification ratio.

(3) The charge carrier conduction in the Au/CuPC/Au and Au/F16CoPC/CuPC/Au devices is described by the space-charge-limited current (SCLC) transport mechanism. However, one can expect the current conduction is contact-limited, rather than the bulk-limited, as the thicknesses of organic films (CuPc and F16CoPC/CuPC) are very small, a few nanometers.

(4) The devices show excellent stability of nearly one month in air, as shown in Fig. S6. The authors attributed to the stable properties of Au electrodes and organic molecules (F16CoPc and CuPc). But the Au electrodes are contaminated with H₂O, NO₂, CO₂, and O₂, as the authors mentioned. Then the contact properties are expected to change over time in the air.

(5) It would be better to show the yield of devices (ratio of successful rectifiers among devices in an array in a chip) and statistical analysis of rectification ratio and 3dB frequencies for devices.

(6) In Fig. 2e "hole accumulation" and "electron accumulation" should be interchanged; i.e., "hole accumulation" in CuPC and "electron accumulation" in F16CoPc.

(7) In line 108, he \diamond The

(8) In Figure S2. d, Relative contents of Co(I), Co(I) and satellite peak ... \diamond ... of Co(I), Co(II) and satellite peak ...

(9) In line 207-208, "our diode" is redundantly used.

Response to the comments of Reviewer 1

Our response:

Thank you very much for your efforts on our manuscript. We greatly appreciate your advices and comments which are very helpful for improving the quality of this work. We have revised the manuscript according to your suggestions. The revised parts are marked by blue font. Our response is presented point-to-point in the following.

Question (1.1) The Title is misleading as 10 MHz is usually considered as relatively low-frequency by the high-frequency electronics community. Therefore, I suggest that "10 MHz" should replace "High-frequency" as for example: "All printed diode operating at 1.6 GHz", PNAS 111, 11943 (2014)

Response (1.1)

Thank you very much for this suggestion. We agree that, although this is the first time to achieve a nanometer-thick integrated organic rectifier with an operation frequency beyond 1 MHz, the obtained maximum 3db operation frequency (~10 MHz) is still in a frequency range considered as “low” by the electronics community. Hence, according to your suggestion, we changed the title into “Integrated Molecular Diode as 10 MHz Half-Wave Rectifier Based on an Organic Nanostructure Heterojunction”.

Question (1.2) There are many ambiguous references to the literature all along the paper.

To give few examples:

- *Abstract: "so far, functional molecular diodes have only been demonstrated in the very low frequency regime". Taking into account Ref.11 ("A 17 GHz molecular rectifier"), "fully integrated molecular diodes" would have been less ambiguous.*

=> Similarly for the sentence "This is the first time a nanometer-thick organic rectifier has been created with operation frequency beyond 1 MHz", and the fact that Ref 11 is omitted from the Fig.5d.

Response (1.2)

We appreciate this comment. It should be pointed out that the claimed 17 GHz molecular rectifier in

Ref.11 is a different type of rectifier compared to our work. In Ref. 11, a Pt tip was biased to the superposition of direct current (DC) and radio frequency (RF) excitation simultaneously (shown in **Figure R1a**). The performance index is the ratio of reflected RF signal power to the incident RF signal power during the DC scan (i.e., S-parameter S_{11})¹. In other words, this kind of rectifier is to rectify the reflected RF signal power which is related to the DC-affected impedance. But the rectifier in our work is a kind of half-wave rectifier, which aims at converting single alternating current (AC) input into the DC output. Please see the comparison table (**Table R1**) below for more information. On the other hand, Nijhuis et al., reported a molecular half-wave rectifier consisting of $\text{Ag}^{\text{TS}}\text{-SC}_{11}\text{Fc//Ga}_2\text{O}_3/\text{EGaIn}^2$, in which they adopted the same molecule (11-ferrocenyl-1-undecanethiol: FcC_{11}SH) with Ref.11, and obtained a high rectification ratio (forward current/reverse current) of 1.0×10^2 at $\pm 1.0 \text{ V}$. However, in their AC-to-DC characterization the devices show a very low retention (18%) of the input voltage amplitude at 50 Hz (shown in **Figure R1b**). This is the reported state-of-the-art molecular scale rectifier device which possesses evident frequency characteristics and can convert AC into DC. It is therefore not proper to compare the result of Ref. 11 with our work in **Figure 5d**, in which all marked results are based on AC-to-DC rectifiers. Moreover, the revised expression “Integrated Molecular Diode as 10 MHz Half-Wave Rectifier” is helpful to distinguish our work from the reported 1) ultra-thin molecular rectifier studies based on probe technologies (e.g., Ref. 11), 2) organic rectifier devices with “thick” active layers (e.g., 100 nm in Ref. 6), and 3) RF rectifier that is used for converting electromagnetic wave (e.g., Ref. 11). A discussion is supplemented on **Page 1** in **Lines 20** and **22**, and **Page 3** in **Lines 59-61** in the manuscript.

[Redacted]

Figure R1. The reported two different types of molecular rectifiers based on 11-ferrocenyl-1-undecanethiol (FcC₁₁SH). **a**, Details about the Au-SC₁₁Fc/Pt tip reported in the 17 GHz molecular rectifier¹. Left panel: experimental setup composed of an interferometer and log current amplifier (resiscope). Right panel: schematic representation of the molecular junction. The Pt tip is biased (through a bias-T) to both DC and RF (1–18 GHz) excitation simultaneously. **b**, Details about the Ag^{TS}-SC₁₁Fc//Ga₂O₃/EGaIn reported in molecular half-wave rectifier². Left panel: the circuit with the molecular junction as the diode in series with an AC signal generator. Right panel: sinusoidal input signals (V_{in} , black, 50 Hz) with the corresponding output signal (V_{out} , red).

Table R1. Comparison between the 17 GHz molecular rectifier¹ and our work.

	17 GHz molecular rectifier^[1]	Our work
What to rectify	Electromagnetic wave	Current (Voltage)
Input	Electromagnetic wave + DC	Alternating current (AC)
Output	Reflected RF signal power	Pulsed DC
Rectification	$\frac{\text{Reflected RF signal power}}{\text{Incident RF signal power}}$	$\frac{\text{Forward current}}{\text{Reverse current}}$

Question (1.3) *The discussion could be improved in many aspects:*

It is usually believed by the molecular electronics community that there is a tradeoff in junction area to optimize the density of current (in nanometer-scale junctions, most of the molecule conduct whereas in tens of micrometer junction, only a fraction of the molecules contribute to the current (W. Du et al Nat. Photonics 10, 274 (2016), and that <3 nm junctions should be the best for high-frequency operation as conductance scales typically exponentially with thickness, but the rectification ratio can be degraded. In that context, the present design choices (10 μm -wide junction and 8 nm thick film) could have been much better discussed, as well as the capacitive network.

To conclude, the paper clearly demonstrates an integrated molecular device operating at 10 MHz with the response of the device to a sine signal. The rolled-up metal microtube approach was surely efficient to reach this goal and DC properties of the molecular diode are nice. However, given the current literature results and the proposed title, a 1 GHz demonstration would have been far more impressive, considering refs 6 and 11.

Response (1.3)

Thank you very much for these comments. In our work, there exists a thickness-related tradeoff between current density and rectification ratio as shown in **Figure R2** (or **Supplementary Fig. 5** in the Supplementary Information). With thinner molecular ensembles (such as 4.5 nm), both forward and reverse currents are high but the rectification ratio at ± 2 V is only about 15. The larger leakage current could make the rectifier invalid to block the reverse current, thus losing the function of rectification. Moreover, the roughness of the top Au tube (~ 2.0 nm) and bottom Au finger (~ 1.3 nm) are in the same range of the molecular film thickness (shown in **Figure R3** or **Supplementary Fig. 6** in the Supplementary Information). The thinner the molecular layer is, the more likely the device gets shorted or breakdown because the effective distance between the two electrodes reduces from the roughness of the Au finger and tube electrodes, leading to a very low yield of successful rectifier devices. On the other hand, for the devices with thicker molecular layer (for instance, 15 nm), the rectification ratio is still maintained as high as about 2 orders, however the forward current is too low to act as an efficient rectifier. Therefore, we chose an 8 nm thick F₁₆CoPc/CuPc hybrid layer to investigate the electrical characteristic, which has not only high rectification ratio but also high

forward current density. We have supplemented the corresponding discussions on **Page 9** in **Lines 206-210** in the manuscript and on **Page 14** in the Supplementary Information (**Supplementary Note 1**).

Figure R2. I-V performance depends on the thickness of the hybrid layer.

Figure R3. Details of the Au (finger)/F₁₆CoPc/CuPc/Au (tube) junction. a, Rolled-up soft-contact demonstrated by AFM image. **b-d**, AFM topography characteristics for rolled-up Au tube (measured from the top surface of the tube), Au finger/F₁₆CoPc/CuPc, and Au finger, respectively. Scale bars, 400 nm. The roughness of the Au tube is extracted from the selected area (in the blue dashed box)

along the lateral direction of the tube. **e**, Schematic illustration of the local contacts. The effective gap between the two electrodes shrinks due to the roughness of the Au finger and the Au tube, as indicated by the blue dashed circle.

The current of the sandwich structure diode increases with the contact area. For our devices, the current could be high enough with a small applied voltage and a narrow mesa width. We designed two kinds of mesas with the nominal designed width (W_{design}), i.e., 5 and 10 μm , and we found that the real widths (W_{real}) of the mesas contacting the tube electrodes are about 1.3 and 7.4 μm , respectively. The shrinking of the designed width is caused by isotropic under-etching of the mesa in HF solution during the micro-fabrication process (shown in **Figure R4** or **Supplementary Fig. 7** in the Supplementary Information). Both of the devices (with $W_{\text{real}} = 1.3 \mu\text{m}$ and $W_{\text{real}} = 7.4 \mu\text{m}$) exhibit good rectification and high current density proportional to their W_{real} , as indicated in **Figure R4**. A discussion is supplemented on **Page 9** in **Lines 210-217** in the manuscript and on **Page 15** in the Supplementary Information (**Supplementary Note 2**).

Figure R4. Relation of current and mesa width. **a**, Schematic of isotropic under-etching of mesa by HF solution. **b** and **c**, scan (upper panels, microscope images) and the corresponding height profiles (lower panels, obtained by height profile meter) of mesas with $W_{\text{design}} = 5$ and $10 \mu\text{m}$. **d**, I-V

characteristics of different diodes based on Au (finger)/F₁₆CoPc (1nm)/CuPc (7 nm)/Au (tube). **e**, Average currents (at 2 V) of devices based on W_{design} = 5 and 10 μm.

In order to investigate the capacitive network, we tried to obtain the detailed equivalent circuit by using impedance spectroscopy (partial data are shown in **Figure R5a** and **b**). However, the impedance data from different measurements on the same device are not consistent, and they could not be fitted well due to rather randomly distributed data points. We think it is caused by the robust but imperfect mechanical contact between the rolled-up tubular electrode and ultrathin molecular ensembles, as illustrated in **Figure R5c**. Owing to the relatively rough surfaces of the CuPc film and Au tube, there are many local tiny contacting areas with ill connection or air gap, resulting in the superposition of many individual parallel RC circuits placed between the CuPc and Au tube electrode (shown in **Figure R5d**). Furthermore, when a DC or AC crosses the device, the involved local high current density and charging/discharging processes might result in local tiny deformation and mechanical vibrations of the tube electrode. As a result, the complicated physical scene and electric process make it difficult to obtain a constant and accurate equivalent circuit in detail. Therefore, we thank you for providing the clue of analysis, and we plan to carry out more detailed investigations and improvements in a future study.

Figure R5. Capacitive network analysis. **a** and **b**, Impedance spectra measured at -0.4 V and -1 V. **c**, Detailed lateral view of the rolled-up soft-contact. **d**, The proposed equivalent electrical circuit. C_1 , C_2 , and C_3 are the capacitances associated with Au (finger)/F₁₆CoPc, F₁₆CoPc/CuPc, and CuPc/Au (tube) interfaces, respectively, with the corresponding shunt resistances R_1 , R_2 , and R_3 . R_s is the series resistance.

Response to the comments of Reviewer 3

Our response:

Thank you very much for your efforts on our manuscript. We greatly appreciate your advices and comments which are very helpful for improving the quality of this work. We have revised the manuscript according to your suggestions. The revised parts are marked by blue font. Our response is presented point-to-point in the following.

***Question (3.1)** In Figure S5a, I-V characteristics of devices with mesa width $W_{\text{mesa}} = 10 \mu\text{m}$ are compared with those with mesa width $W_{\text{mesa}} = 5 \mu\text{m}$. Since mesa width difference is 2, the current difference should be around 2 times if the top contact created by a rolled-up Au microtube is similar. However, it appears that currents are at least 4 or 5 times higher for devices with mesa width $W_{\text{mesa}} = 10 \mu\text{m}$ compared to $W_{\text{mesa}} = 5 \mu\text{m}$. This may suggest that there exists a lateral leakage current. Then it seems that better molecular scale rectifier can be achieved for devices with smaller mesa width. Analyze the effect of lateral leakage current on the rectifier performance.*

Response (3.1)

Thank you very much for your comments. In the device fabrication, the designed mesa width (W_{design}) was defined by the lithography patterns. However, the real width (W_{real}) is influenced by the isotropic under-etching by HF solution during the mesa formation (shown in **Figure R6a** or **Supplementary Fig. 7a** in the Supplementary Information). During HF etching, SiO_2 is removed uniformly from all available directions, resulting in a shrunken top part of the mesa when compared to the designed size (W_{design}). As a result, the real mesa widths (W_{real}) of the designed $W_{\text{design}} = 5$ and $10 \mu\text{m}$ are about 1.3 and $7.4 \mu\text{m}$, respectively (shown in **Figure R6b-c** or **Supplementary Fig. 7b-c** in the Supplementary Information). Then the real width ratio (noted as Ratio 1) corresponding to ($W_{\text{design}} = 10 \mu\text{m} / W_{\text{design}} = 5 \mu\text{m}$) is 5.69 and not 2 . On the other hand, as shown in **Figure R6d-e** (**Supplementary Fig. 7d-e** in the Supplementary Information), the average current ratio of devices based on $W_{\text{design}} = 10 \mu\text{m}$ over $W_{\text{design}} = 5 \mu\text{m}$ is calculated to be 5.36 (noted as Ratio 2), close to Ratio 1. Therefore, the forward current can be considered as proportional to the real mesa width, and a lateral leakage current can be safely neglected. To avoid the ambiguity in the mesa width, in the

revised manuscript we use W_{design} to replace the previous W_{mesa} . A discussion is supplemented in the manuscript on **Page 9** in **Lines 210-216** and in the Supplementary Information on **Page 15** (**Supplementary Note 2**).

Figure R6. Relation of current and mesa width. **a**, Schematic of isotropic under-etching of mesa by HF solution. **b** and **c**, scan (upper panels, microscope images) and the corresponding height profiles (lower panels, obtained by height profile meter) of mesas with $W_{\text{design}} = 5$ and $10 \mu\text{m}$. **d**, I-V characteristics of different diodes based on Au (finger)/F₁₆CoPc (1nm)/CuPc (7 nm)/Au (tube). **e**, Average currents (at 2 V) of devices based on $W_{\text{design}} = 5$ and $10 \mu\text{m}$.

Question (3.2) As the authors mentioned and depicted in Fig. S5b, the real electrode contact area for the device with mesa width $W_{\text{mesa}} = 10 \mu\text{m}$ should be much less than 100 m^2 . Then a forward current density can be much higher than $\sim 200 \text{ A/cm}^2$ at +2 V. Under such high current across very thin ($\sim 8\text{nm}$) organic layer, one can expect quite high Joule heating, resulting in increased junction temperature. And the effect of Joule heating and junction-temperature rise should depend on the width of mesa. Since the charge carrier injection and conduction depend on the temperature, it is necessary to address the effect of Joule heating on the device I-V characteristics and the rectification

ratio.

Response (3.2)

Thank you very much for the comment. It is challenging to avoid the critical Joule heating effect of molecular-scale electronics. As shown in **Figure R7a-b (Supplementary Fig. 9a-b** in the Supplementary Information), for the devices in this work, during a 40-cycle IV measurement, the rectification ratio (RR) decreased from 270 to 230, and the forward current at 2V maintained at a high-level current of more than 200 μA . The decrease in rectification is considered to be caused by Joule heating which might increase the currents. However, the devices still work under such high current density due to the good thermal stability of phthalocyanine materials. Our experiments prove that there is a relatively safe current density region, above which the device tends to suffer burnout due to the excessively accumulated heat. As shown in **Figure R7d-e (Supplementary Fig. 9d-e** in the Supplementary Information), when the current density is as high as ca. 1200 A/cm^2 at 2V (taking the contact area as 74 μm^2), the device suddenly broke down during multiple bias scanning from positive to negative voltage. As a result, the device lost its rectifying function. Furthermore, Joule heating can also destroy the rectifying devices during the frequency measurements, as shown in **Figure R7f or Supplementary Fig. 9f** in the Supplementary Information. The corresponding discussion is supplemented in the revised manuscript on **Page 9** in **Lines 223-224** and in the Supplementary Information on **Page 16 (Supplementary Note 3)**.

Figure R7. Stability of the devices based on Au (finger)/F₁₆CoPc (1nm)/CuPc (7 nm)/Au (tube), with $W_{\text{design}} = 10 \mu\text{m}$. **a**, Typical I-V characteristics. **b**, corresponding device cycling performance: cycling number dependent current at 2V and rectification ratio (RR). **c**, I-V characteristics over 420 days. **d**, Device breakdown during the I-V measurement caused by Joule heating. **e**, I-V characteristics of the device in **d** after breakdown. **f**, Demonstration of device breakdown during the frequency measurement.

Question (3.3) The charge carrier conduction in the Au/CuPc/Au and Au/F₁₆CoPc/CuPc/Au devices is described by the space-charge-limited current (SCLC) transport mechanism. However, one can expect the current conduction is contact-limited, rather than the bulk-limited, as the thicknesses of organic films (CuPc and F₁₆CoPc/CuPc) are very small, a few nanometers.

Response (3.3)

Thank you very much for this comment. The CuPc/Au (tube) interface formed by the robust mechanical contact can block the hole injection from the Au tube into the molecular layer, however, the hole crossing from the molecular layer to the Au tube is not blocked. This interface is mainly responsible for the function of rectification, as demonstrated by the I-V characteristics (**Figure 4a**). In other words, under forward bias condition the charge transport (from Au finger to Au tube electrode) is not limited when compared to the reverse bias condition (from Au tube to Au finger

electrode). To further clarify the charge transport process in forward direction, apart from the SCLC model³, the forward-direction currents are also fitted with two possible contact-limited conduction models (i.e., Fowler-Nordheim tunneling and Schottky emission⁴), as shown in **Figure R8** (**Supplementary Fig. 10** in the Supplementary Information). As we can see, only the SCLC model fits well, which exhibits the typical three transport regions with noticeable different slopes (shown in **Figure R8a**): Ohmic transport ($m \approx 1$, Regime I), shallow trap-limited SCLC ($m \approx 2$, Regime II), and deep trap-filled limit conduction ($m > 2$, Regime III)^{5, 6}. This indicates, the current under forward bias complies to the trap-controlled SCLC mechanism, implying that the forward-direction transport is most likely bulk-limited, although the thickness of the molecular layer is only a few nanometers. Furthermore, the high-density traps (about $1.58 \times 10^{18} \text{ cm}^{-3}$ for water-treated CuPc) makes bulk-limited conduction possible to happen because of its high bulk resistance. A discussion is supplemented on **Page 11** in **Lines 261-266** in the manuscript and on **Page 19** in the Supplementary Information (**Supplementary Note 5**).

Figure R8. Conduction model fitting for the forward I-V data. **a**, Space-charge-limited current model (bulk-limited). **b**, Fowler-Nordheim tunneling model (contact-limited) is applied to the forward-direction current. The right panel shows the region in the dash box of the left panel. **c**, Schottky emission model (contact-limited) is applied to the forward-direction current.

Question (3.4) The devices show excellent stability of nearly one month in air, as shown in Fig. S6.

The authors attributed to the stable properties of Au electrodes and organic molecules ($F_{16}CoPc$ and $CuPc$). But the Au electrodes are contaminated with H_2O , NO_2 , CO_2 , and O_2 , as the authors mentioned. Then the contact properties are expected to change over time in the air.

Response (3.4)

Thank you very much for this comment. Considering the practical scenario of the device formation, the contamination with H_2O , NO_2 , CO_2 , and O_2 , happened to the Au electrodes and molecular layers during the multi-step fabrication process, which was revealed by the UPS and XPS results. We think once the devices are formed, a certain influence of further contamination in the stable environment atmosphere will still take place, but will not be significant enough to dramatically change the device performance over a short time. To demonstrate the change of contact properties over a long time in air, after 420 days since the first measurement, the I-V characteristics of a device is measured again. It is found that both the forward and reverse currents increased slightly, while the rectification ratio decreased a little, as shown in **Figure R9 (Supplementary Fig. 9c** in the Supplementary Information). This may be ascribed to the oxygen doping⁷ and/or further contamination happening to the device⁸. A discussion is supplemented in the revised manuscript on **Page 9 in Lines 223-224** and on **Page 16** in the Supplementary Information (**Supplementary Note 3**).

Figure R9. Stability of I-V performance over 420 days.

Question (3.5) It would be better to show the yield of devices (ratio of successful rectifiers among devices in an array in a chip) and statistical analysis of rectification ratio and 3dB frequencies for devices.

Response (3.5)

Thank you very much for this comment. The construction of our rectifiers is based on the rolled-up nanomembrane which contacts the ultrathin molecular ensembles from the top, thus providing a damage-free and self-adjusted electrode. Any defects in sacrificial layer or strain layer could lead to the failure of rolling. For a 9×6 device array, the yield of initial devices with successful rolled-up contacts is about 80%, as shown in **Figure R10a (Supplementary Fig. 12a** in the Supplementary Information). Among the rolled-up devices, some are open circuit due to the disconnection of the rolled-up tubes from the circuits on the substrate, and some are short circuit. As mentioned before, the film thickness of the F₁₆CoPc/CuPc hybrid layer in the demonstrating devices is about 8 nm (shown in **Figure 3d** in the manuscript). However, the roughness of the top Au tube, F₁₆CoPc/CuPc hybrid layer and bottom Au finger are approximately 2.0, 2.3 and 1.3 nm (shown in **Figure R11 b-d** or **Supplementary Fig. 6b-d** in the Supplementary Information), respectively, which are in the same range with the organic layer thickness. This could shrink the effective gap between the two electrodes sandwiching the F₁₆CoPc/CuPc ultrathin layer (shown in **Figure R11e** or **Supplementary Fig. 6e** in the Supplementary Information), resulting in the short-circuit or easy breakdown. At last, about 44% (24 devices) can be obtained as successful diodes (shown in **Figure R10a** or **Supplementary Fig. 12a** in the Supplementary Information) and 13% (7 devices) possess evident rectification at the frequencies larger than 100 kHz (**Figure R10b**). For the surviving rectifiers, as shown in **Figure R10b** the devices achieve a maximum 3db frequency of more than 10 MHz and an average 3dB frequency of 3.81 MHz. A discussion is supplemented on **Page 14** in **Lines 343-344** in the manuscript and on **Page 20** in the Supplementary Information (**Supplementary Note 6**).

Figure R10. Statistical analysis of the rectifier array on a chip. a, Statistics of the 54 fabricated devices. RR= rectification ratio. **b**, Frequency performance of rectifiers.

Figure R11. Details of the Au (finger)/F₁₆CoPc/CuPc/Au (tube) junction. a, Rolled-up soft-contact demonstrated by AFM image. **b-d**, AFM topography characteristics for rolled-up Au tube (measured from the top surface of the tube), Au finger/F₁₆CoPc/CuPc, and Au finger, respectively. Scale bars, 400 nm. The roughness of the Au tube is extracted from the selected area (in the blue dashed box) along the lateral direction of the tube. **e**, Schematic illustration of the local contacts. The effective gap between the two electrodes shrinks due to the roughness of the Au finger and the Au tube, as indicated by the blue dashed circle.

Question (3.6) In Fig. 2e “hole accumulation” and “electron accumulation” should be interchanged; i.e., “hole accumulation” in CuPC and “electron accumulation” in F₁₆CoPc.

Response (3.6)

Thank you very much for your comments and suggestions. The figure has been corrected, as shown in the updated **Figure 2e** in the manuscript.

Question (3.7) In line 108, he →The

Response (3.7)

“he CuPc” is corrected to “The CuPc” in the revised manuscript on **Page 5** in **Line 105**.

Question (3.8) In Figure S2. d, Relative contents of Co(I), Co(I) and satellite peak ... → ... of Co(I), Co(II) and satellite peak ...

Response (3.8)

“Relative contents of Co(I), Co(I)” is corrected to “Relative contents of Co(I), Co(II)” in the revised Supplementary Information on **Page 3**.

Question (3.9) In line 207-208, “our diode” is redundantly used.

Response (3.9)

The redundantly used “our diode” is corrected in the revised manuscript on **Page 9** in **Lines 217-219**.

References

1. Trasobares, J., Vuillaume, D., Théron, D. & Clément, N. A 17 GHz molecular rectifier. *Nat. Commun.* **7**, 12850 (2016).
2. Nijhuis, C. A., Reus, W. F., Siegel, A. C. & Whitesides, G. M. A molecular half-wave rectifier. *J. Am. Chem. Soc.* **133**, 15397-411 (2011).
3. Xu, G. et al. Bulk-like electrical properties induced by contact-limited charge transport in organic diodes: revised space charge limited current. *Adv. Electron. Mater.* **4**, 1700493 (2018).
4. Chiu, F. C. A review on conduction mechanisms in dielectric films. *Adv. Mater. Sci. Eng.* **2014** (2014).
5. Montero, José M. et al. Trap-limited mobility in space-charge limited current in organic layers. *Org. Electron.* **10**, 305-312 (2009).
6. Samanta S. et al. Understanding of multi-level resistive switching mechanism in GeO_x through redox reaction in H₂O₂/sarcosine prostate cancer biomarker detection. *Sci. Rep.* **7**, 1-12 (2017).
7. Anthopoulos T., Shafai T. Oxygen induced p-doping of α -nickel phthalocyanine vacuum sublimed films: Implication for its use in organic photovoltaics. *Appl. Phys. Lett.* **82**, 1628-1630 (2003).
8. Grobosch, M. & Knupfer, M. Charge-injection barriers at realistic metal/organic interfaces: metals become faceless. *Adv. Mater.* **19**, 754-756 (2007).

REVIEWERS' COMMENTS:

Reviewer #1 (Remarks to the Author):

I appreciate the efforts made by the authors that include a clearer title, abstract, additional data/discussion on the film thickness and the fair comments on the capacitive network. I believe that the paper deserves to be published in Nature Communications as it can be used as a clear performance reference for future studies from both molecular electronics and organic electronics fields. Overall, it is a nice piece of work.

I still suggest some minor modifications:

* The Intro is still not clear enough, at least from a broad audience perspective.

- For example, it is mentioned that such devices could operate up to THz range, but it is not clearly mentioned what demonstrations have been obtained so far towards this target, and their limitations. One important reference is "Quantum Plasmon Resonances Controlled by Molecular Tunnel Junctions" Science, 343, 6178 (2014), as it is shown that quantum tunneling through molecules is clearly operational in the range 140-240 THz. This demonstration does not include the "rectification function" though.

Ref 11 demonstrates that the rectification function is still working until 17 GHz, but not in an integrated device configuration. [comment to the difference in current and S11: authors suggest in their answer to referee that current and S11 are very different, but in fact the experimental setup using alternating current in this study is limited to relatively low frequency. If the authors want to make their device operate in the GHz range or higher in the future, they will likely have to consider the S parameters approach.

One ref from P.S Weiss with is AC-STM on molecules (GHz) would have been fair as well.

- Related: In the intro, the sentence "However, these techniques are unable to exploit the intrinsic nanoscale properties ..."; this sentence is too aggressive or ambiguous.

As a matter of fact, the Nature communication paper "Controlling the direction of rectification in a molecular diode" by Nijuis et al. nicely exploits intrinsic nanoscale properties, with an EGaIn approach.

Maybe the authors wanted to mention: Large area techniques including GaIn tend to reduce the electronic coupling to the upper part of the junction, which reduces frequency range operation. Therefore, while a GHz range molecular diode has been demonstrated using an AFM based technique, one important goal is to demonstrate functional molecular rectifiers operating at high frequency using an integrated top electrode.

* The capacitive network => I understand that precise estimation is difficult, but just a // plate configuration, taking a dielectric constant of 2.2, the thickness and junction area would give a rough approximation for the capacitance, and the theoretical cut-off frequency taking into consideration the capacitances.

* In the discussion, just a prediction of the theoretical limitation for this current device and future perspective/suggestions for the future devices is still missing. For example, in "Molecular diodes, Breaking the Landauer limit", Nature Nano 12, 725(2017), it is mentioned that we have to consider some tradeoff between rectification and high-frequency operation (if the electronic coupling is large, frequency operation is large, but rectification ratio is degraded; this is the opposite for weak coupling).

The device proposed by the authors is a good example of nice tradeoff (good rectification ratio and high frequency of operation) => it is worth mentioning.

Response to the comments of Reviewer 1

Our response:

Thank you very much for your efforts on our manuscript. We greatly appreciate your affirmation and encouragement on our work, and your advice is very helpful for improving the quality of this work. We have revised the manuscript according to your suggestions. The revised parts are marked by blue font. Our response is presented point-to-point in the following.

Question (1.1) *The Intro is still not clear enough, at least from a broad audience perspective.*

- For example, it is mentioned that such devices could operate up to THz range, but it is not clearly mentioned what demonstrations have been obtained so far towards this target, and their limitations.

One important reference is "Quantum Plasmon Resonances Controlled by Molecular Tunnel Junctions" Science, 343, 6178 (2014), as it is shown that quantum tunneling through molecules is clearly operational in the range 140-240 THz. This demonstration does not include the "rectification function" though.

Ref 11 demonstrates that the rectification function is still working until 17 GHz, but not in an integrated device configuration. [comment to the difference in current and S11:

authors suggest in their answer to referee that current and S11 are very different, but in fact the experimental setup using alternating current in this study is limited to relatively low frequency. If the authors want to make their device operate in the GHz range or higher in the future, they will likely have to consider the S parameters approach.

One ref from P.S Weiss with is AC-STM on molecules (GHz) would have been fair as well.

- Related: In the intro, the sentence "However, these techniques are unable to exploit the intrinsic nanoscale properties ..."; this sentence is too aggressive or ambiguous.

As a matter of fact, the Nature communication paper "Controlling the direction of rectification in a molecular diode" by Nijuis et al. nicely exploits intrinsic nanoscale properties, with an EGaIn approach.

Maybe the authors wanted to mention: Large area techniques including GaIn tend to reduce the electronic coupling to the upper part of the junction, which reduces frequency range operation.

Therefore, while a GHz range molecular diode has been demonstrated using an AFM based technique, one important goal is to demonstrate functional molecular rectifiers operating at high frequency using an integrated top electrode.

Response (1.1)

-Thank you very much for these comments, especially for the suggestion about the S parameter approach which provides us with a valuable clue in future. In order to improve the introduction part and include the previous efforts made to address the THz target in the molecular electronics field, we revised the manuscript (on **Page 2** in **Lines 43-50**).

-Thank you for pointing out the impropriety of the sentence "However, these techniques are unable to exploit the intrinsic nanoscale properties ...", we revised the manuscript (on **Page 3** in **Lines 59-65**).

***Question (1.2)** The capacitive network => I understand that precise estimation is difficult, but just a // plate configuration, taking a dielectric constant of 2.2, the thickness and junction area would give a rough approximation for the capacitance, and the theoretical cut-off frequency taking into consideration the capacitances.*

Response (1.2)

We appreciate this comment. According to your suggestion, we calculated the approximate capacitance and the theoretical cut-off frequency of our molecular diode, and revised the manuscript (on **Page 15** in **Lines 366-374**). Here, we assume a dielectric constant of 4 for phthalocyanine materials, as this has been reported in previous studies, for instance, Ref. 52 and 53.

***Question (1.3)** In the discussion, just a prediction of the theoretical limitation for this current device and future perspective/suggestions for the future devices is still missing.*

For example, in "Molecular diodes, Breaking the Landauer limit", Nature Nano 12, 725(2017), it is mentioned that we have to consider some tradeoff between rectification and high-frequency

operation (if the electronic coupling is large, frequency operation is large, but rectification ratio is degraded; this is the opposite for weak coupling).

Response (1.3)

Thank you very much for this suggestion. In the Discussion part, through analyzing the related parameters, we provide some opinions and perspectives for the future devices, and revised the manuscript (on **Page 15** in **Lines 365-378** and **Page 16** in **Lines 389-392**).